# How Parenting and Family Characteristics Predict the Use of Feeding Practices among Parents of Preschoolers: A Cross-Sectional Study in Beijing, China

**DOI:** 10.3390/nu14153109

**Published:** 2022-07-28

**Authors:** Denghui Hu, Yuxiang Tang, Lutong Zheng, Kaiyuan Min, Fenghua Su, Jing Wang, Wei Liao, Ruijie Yan, Yueqing Wang, Xiaoyan Li, Juan Zhang

**Affiliations:** 1School of Population Medicine and Public Health, Peking Union Medical College, Chinese Academy of Medical Sciences, Beijing 100730, China; hdh@student.pumc.edu.cn (D.H.); tyxxkai@163.com (Y.T.); yanrj@sph.pumc.edu.cn (R.Y.); wyqv-d@hotmail.com (Y.W.); 2Center of Social Medicine, National Research Institute for Family Planning, Beijing 100081, China; 3School of Health Humanities, Peking University, Beijing 100191, China; zhenglt@pku.edu.cn; 4Institute of Basic Medical Sciences, Peking Union Medical College, Chinese Academy of Medical Sciences, Beijing 100730, China; min_kaiyuan@163.com (K.M.); liaowei@pumc.edu.cn (W.L.); 5Dongcheng Center for Disease Control and Prevention, Beijing 100013, China; sufh0529@163.com (F.S.); jing2000@sohu.com (J.W.); 6Faculty of Psychology, Beijing Normal University, Beijing 100091, China; xiaoyan.li@mail.bnu.edu.cn

**Keywords:** preschoolers, feeding practices, weight perceptions, parenting, only child

## Abstract

Parental coercive and structured feeding practices are linked with children’s weight gain. Thus, identifying their predictors will assist in childhood obesity prevention. We explored how parents’ concerns and perceptions of children’s weight, parenting stress, parenting competence, parents’ family roles, and only child status (of both parent and child) predict the use of restriction, pressure to eat, and monitoring practices among parents of preschoolers. Parent–child dyads (*n* = 2990) were recruited in Beijing in 2019. Parenting competence, parents’ weight perceptions and feeding practices were assessed using the Chinese version of Parenting Sense of Competence Scale and Child Feeding Questionnaire (CFQ), respectively. Parenting stress and other variables were collected through self-administered questionnaires. Multivariate linear associations between parents’ weight perceptions and feeding practices were significant among normal-weight children. Parents’ concerns about children being overweight were positively associated with restriction and monitoring, and negatively associated with pressure to eat. Higher levels of parenting stress and parenting competence significantly improved the adoption of restriction and pressure to eat. Parents’ only child status and that of children had an impact on parents’ feeding practices. The fathers’ feeding preferences were substantially different from what mothers preferred. In conclusion, such parenting and family characteristics significantly influenced feeding practices of preschoolers’ parents. These were long neglected in China.

## 1. Introduction

The world has witnessed a rapid increase in the prevalence of overweight and obesity among preschoolers [1]. Similarly, in China, overweight and obesity have increased rapidly in the past four decades. The latest national prevalence estimates were 10.4% for overweight or obesity in children younger than 6 years by the China Chronic Disease and Nutrition Surveillance 2015–2019 survey [2]. Children who are overweight and obese during early childhood are likely to stay obese into adulthood and develop diseases such as type 2 diabetes, cardiovascular disease, and cancer [3,4].

Although the causes of overweight or obesity are intricate, children’s unhealthy eating behavior is one of the remarkable factors [5,6,7]. Previous studies have indicated that appropriate parental feeding practices are constructive for shaping children’s healthy eating behavior and managing their weight status [8,9,10,11]. However, most parents, actually, often misunderstand their children’s development of normal eating behaviors, then respond with coercive and structured feeding practices, such as restriction, pressure to eat, and monitoring [12]. Such feeding practices are counterproductive since they may cause children to eat for emotional or stress-related reasons instead of appetite, then resulting in extra energy consumption [13,14,15,16]. Amber et al. [17] also concluded a common feature that coercion is a parent-centered strategy which serves the parents’ goals and desires without considering the children’s needs. Therefore, detailing the contributing factors of parental coercive and structured feeding practices helps inform younger children and prevent obesity.

Abundant studies have reported that the perceptions of a child’s weight status may influence the parents’ adoption of feeding practices [18,19]. For example, parents who believe that their child is overweight report greater restriction on their child’s diet [20,21]. A systematic review concluded that parents often misunderstand children’s weight status, especially in preschoolers [22]. Additionally, in China, a high proportion of parents often misinterpret children’s large body weight and size as healthy growth [2]. A previous study also reported that parents whose children have higher body mass index (BMI) Z-scores express concerns about their children being overweight or obese, which ultimately affected their feeding practices, such as restriction [19]. Given this evidence, parent–child feeding interactions, whilst considering the parents’ perceptions and concerns about preschoolers’ weights, deserve further investigation in China.

Existing evidence indicates that parenting psychological predictors, especially parenting stress and parenting competence (including dimensions of self-efficacy, skill, communication, etc.) might influence the parents’ adoption of feeding practices [23,24]. This kind of stress was defined as the perceived degree of stress when rearing their children, and parenting competence is the cognitive aspect of parenting [25]. However, previous studies have revealed conflicting findings regarding the relationships between parenting and feeding practices [26,27,28,29]. Some have shown that higher parenting self-efficacy is associated with less adoption of pressure to eat [27]. However, a study on UK mothers of 18- to 59-month-old children pointed out no significant association between parenting self-efficacy and pressure to eat [29]. Thus, further studies are needed to elucidate the relationships and understand how such predictors might be related to Chinese parents’ coercive and structured feeding practices. 

Additionally, due to the family planning policy implementation in China, the single child used to be overfed by parents [30]. However, as China has recently ended its one-child policy, the Chinese family structure may change, especially regarding the number of children’s siblings. Previous studies have indicated that only child status and number of siblings may influence the parent–child feeding interactions among younger Western children [31]. Therefore, it is necessary to understand the relationships between only child status (of both parents and children) and parental feeding practices in China.

Above all, this study examined the relationships between parents’ perceptions and concerns about children’s weight status, parenting stress, parenting competence, parents’ family roles, only child status (of both parent and child), and parental feeding practices among Chinese preschoolers and one of their parents. This study is expected to help understand the influence of parenting and family characteristics on the parental adoption of feeding practices and then provide guidance on future studies regarding parent–child feeding interactions and childhood weight management.

## 2. Materials and Methods

### 2.1. Study Design and Procedures

#### 2.1.1. Study Design and Sampling

This cross-sectional study was conducted in local kindergartens in the Dongcheng District, the eastern half of the downtown area of Beijing, China. This study’s design and sampling methods have been described elsewhere in detail [32]. Briefly, we used a stratified proportionate cluster sampling method to select the parent–child dyads. Taking each class as a cluster, we calculated the sample size according to the formula $n=k\times \frac{u_{\alpha }^{2}P\left(1-P\right)}{\delta^{2}}$. Then, estimated through a pilot study, the number of parent–child dyads needed was calculated to be 3592. Eligible participants were children registering in the second or third year of the chosen kindergartens and one of their parents. 

#### 2.1.2. Procedures

With the written informed consent of preschoolers and their legal parents, all 3595 children and one of their parents were invited to take part in the study from April to June 2019. The electronic questionnaires were distributed by teachers in charge of the classes with unique ID numbers. One of children’s parents completed them anonymously. One week later, questionnaires were collected and preliminarily checked by teachers in charge of the classes.

This survey invited a total of 3595 parent–child dyads, and 3057 parent–child dyads participated, with a response rate of 85% (3057/3595). After excluding 67 (2.2% of total) non-parent–child relationship dyads with incomplete information, 2990 parent–child dyads were included in the final analysis. The inclusion and exclusion process of the study population is shown in Figure 1.

### 2.2. Measurement

#### 2.2.1. Parental Feeding Practices

The Chinese version of the Child Feeding Questionnaire was adapted from the widely used Child Feeding Questionnaire (CFQ) developed by Birch and Fisher [33] to account for the Chinese context. The questionnaire consists of 7 subscales to assess parental beliefs, attitudes, and practices regarding child feeding. Three parental coercive and structured feeding practice subscales (including 15 items and assessing restriction, pressure to eat, and monitoring practices) were derived from parents’ self-reported responses to the adapted Chinese version of CFQ. The 15-item questionnaire assesses parental feeding practices using a 5-Likert scale (disagree, slightly disagree, neutral, slightly agree, agree). Participant scores for each feeding practice subscale were calculated as the mean of all available items. A higher score indicated more adaptation to the feeding practice. Cronbach’s α of the three feeding practices was 0.78, 0.70, and 0.95, respectively.

#### 2.2.2. Parenting

Parenting practices, including parents’ weight perceptions, parents’ concerns about their child being overweight, parenting stress, and parenting competence, were captured by scales adapted from the widely used questionnaires or developed by authors based on previous literature and work in China.

Parents’ weight perceptions were measured by a single question developed from the CFQ [33]: “How would you describe your child’s weight currently?” with response options of “very underweight”, “underweight”, “normal weight”, “overweight”, and “very overweight”. The response options were recorded into underweight (“very underweight” and “underweight”), normal weight, and overweight (“overweight” and “very overweight”) groups. We compared parents’ response options with the children’s actual BMI status assessed by the school nurse. Parents’ weight perceptions were categorized as misperception (underestimation and overestimation) group and correct perception group.

Concerns about their child being overweight were measured using a single item from CFQ: “how concerned are you about your child becoming overweight?”. The response options were recorded into concerned and unconcerned groups.

Six items of parenting stress, which asked for the respondent’s agreement on a 5-Likert scale to the following statements, were recorded: “I feel that I fail to be a good caregiver”; “I face many stresses to raise and educate the child”; “Raising the child is more difficult than I imagine”; “I have financial burden to take care of the child”; “Taking care of the child brings many problems in my life”; and “I have no energy to take care of the child because of my health status”. 

For parenting competence, we selected and adapted some items from the Parenting Sense of Competence Scale based on the parents’ local contexts and educational levels [34]. The four items of parenting competence also asked the parents’ agreement on a 5-Likert scale to the following statements: “I am capable of taking care of and guiding the child”; “I believe I have the skills necessary to be a good caregiver”; “I know how to communicate with him/her”; and “I am a good problem solver no matter what happened to him/her”. In the current study, a mean score of all items was calculated as the factor scores of parenting stress and parenting competence, respectively, and Cronbach’s α was 0.86 and 0.92. Higher scores indicated higher levels of parenting stress and parenting competence.

#### 2.2.3. Family Characteristics and Children’s BMI

Child age, gender, parental family roles, parental marital status, household income per capita per year, and only child status of both children and parents were collected via single questions, respectively.

The trained school nurses measured the children’s weights and heights annually following standardized anthropometry measurement protocols developed by the Child Care Center of Beijing Health Bureau to the nearest 0.1 kg and 0.1 cm. The children’s weight and height data were attached to unique ID numbers matching their questionnaire data. We calculated the children’s body mass index (BMI) values and defined weight status using criteria developed by the World Health Organization (WHO) [35,36,37].

### 2.3. Ethical Consideration

The Ethics Committee of Dongcheng Center for Disease Control and Prevention approved the study (DCCDPCIRB-20180416-1). All respondents gave their informed consent for inclusion before participating in the study.

### 2.4. Statistical Analysis

The database was established through WJX (a free Chinese platform providing functions of online survey, Changsha, China: Changsha Ranxing Information Technology Co., Ltd.) and Microsoft Excel 2019 (Microsoft Corp, Redmond, WA, USA); statistical analysis was conducted with R-studio Version 1.1.383 (1999 Free Software Foundation, Boston, Massachusetts, MA, USA: RStudio, PBC). Continuous and categorical variables were presented as means (SDs) and frequencies (percentages), and the medians (mix, max) were also presented for the continuous variables. First, after testing the homogeneity of variance, Welch’s analysis of variance (ANOVA), one-way ANOVA, and Kruskal–Wallis rank sum tests were, respectively, applied to investigate the differences, and post hoc analyses were conducted to analyze the differences between statistically significant groups. Then, multivariate linear regression models were applied to ascertain the associations between parenting, family characteristics, and parental feeding practices. Regression models for the parents’ perceptions and concerns about their children’s weights were stratified by the children’s actual weight status, and all models were adjusted for child age, gender, parents’ marital status, and household income. Regression coefficients (*β*) and 95% confidence intervals (95%CI) were identified. A two-tailed *p* < 0.05 was considered statistically significant in all analyses. 

## 3. Results

### 3.1. Sample Characteristics

Table 1 presents the demographic characteristics of the study participants. Among 2990 preschoolers, the majority were boys (51.7%), and the mean (SD) age was 66.3 (7.3) months (5.5 years old). The mean (SD) BMI was 15.6 (1.8) and the prevalence of overweight and obesity was 12.1% and 5.7%, respectively, for the preschoolers. Over two-thirds (68.9%) of preschoolers were an only child. The majority (77.0%) of the paired parents were mothers, and 44.6% were both from one-child families. About 95.4% of the paired parents were married. Only 8.7% of the family’s household income per capita per year was less than CNY 80,000 (equal to USD 11,928). 

### 3.2. Scores of Feeding Practices

Table 2 shows the mean (SD) scores of the feeding practices by the children’s actual weight status. The mean (SD) scores of restriction, pressure to eat, and monitoring practices were 3.8 (0.8), 3.2 (0.8), and 3.9 (1.0) out of a possible score of 5, respectively, suggesting moderate to severe adoption of such feeding practices. After conducting Levene’s test for the equality of variances, two Welch’s ANOVA tests (for heteroskedasticity groups) and a one-way ANOVA test were applied between restriction, pressure to eat, monitoring, and children’s weight status, respectively. Preschoolers who received higher levels of restriction and lower levels of pressure to eat appeared to have higher levels of current weight status (Welch’s ANOVA *F* = 8.8, 21.1, *p* < 0.001, 0.001, respectively). 

### 3.3. Parenting

#### 3.3.1. Parents’ Perceptions and Concerns about Weight

Only 11.8% of parents perceived their children’s weight status as overweight or obese, but 17.8% of the children’s weight status was indeed overweight or obese based on measured height and weight. Additionally, only 77.4% (*n* = 2315) of parents correctly assessed the children’s weight status (see Table 1). Figure 2 shows the parents’ perceptions of children’s weight status and concerns about their child being overweight. Misperceptions of children’s weights widely existed in each group. More than one-third of parents of underweight and overweight children misestimated their children’s weight status. The heavier the child in regard to their weight status, the higher the proportion of parents underestimating the children’s body weight (Figure 2 left. Kruskal–Wallis $\chi^{2}$ = 281.8, *p* < 0.01), despite the higher proportion of parents expressing concern about their child being overweight (Figure 2 right. Kruskal–Wallis $\chi^{2}$ = 238.41, *p* < 0.01).

The results of the linear regression models of parental perceptions of children’s weights, concerns about their child being overweight, restriction, pressure to eat, and monitoring practices are shown in Table 3. Stratified by children’s actual weight status, associations between parental weight misperceptions and feeding practices were only significant in children with normal weights. Parents who perceived their children as underweight were more likely to pressure them to eat (*β* = 0.333, 95%CI: (0.243, 0.423)), and those who overestimated their children’s weights were less likely to use pressure eating practices (*β* = −0.272, 95%CI: (−0.518, −0.026)). Parents’ concerns about their child being overweight were positively associated with restriction and monitoring both in normal-weight (*β* = 0.207, 0.129, 95%CI: (0.140, 0.273), (0.046, 0.213), respectively) and overweight children (*β* = 0.501, 0.356, 95%CI: (0.316, 0.686), (0.102, 0.611), respectively), and negatively associated with pressure to eat in normal-weight children (*β* = −0.078, 95%CI: (−0.145, −0.011)). 

#### 3.3.2. Parenting Stress and Competence

The mean (SD) score for parenting stress was 2.3 (0.8), out of a possible score of 5, and that of parenting competence was 4.0 (0.6), out of a possible score of 5, indicating mild to moderate parenting performance (see Table 1). 

The linear regression coefficients for the parenting stress and parenting competence of three feeding practices are shown in Table 4. Parenting stress was positively associated with parental restriction practice (*β* = 0.048, 95%CI: (0.010, 0.087)) and pressure to eat (*β* = 0.159, 95%CI: (0.120, 0.198)), showing no statistically significant association with monitoring practice. Parenting competence was significantly associated with restriction, pressure to eat, and monitoring feeding practices (*β* = 0.057, 0.060, 0.253, 95%CI: (0.084, 0.105), (0.010, 0.109), (0.193, 0.314), respectively). 

### 3.4. Family Characteristics

The linear regression analysis results of the only child status of both parents and preschoolers and family roles are shown in Table 4. Compared with the only child children, the non-only children’s parents were more likely to adopt pressure to eat (*β* = 0.108, 95%CI: (0.044, 0.171)). Compared with children whose mother and father both were not from a one-child family, those who had one or two parents from the one-child family were less likely to adopt a restriction practice (*β* = −0.109, −0.096, 95%CI: (−0.200, −0.019), (−0.171, −0.020), respectively). Mothers from one-child families were less likely to adopt pressure to eat practices (*β* = −0.097, 95%CI: (−0.200, −0.019)). Parents who both came from one-child families were more likely to adopt monitoring practices (*β* = 0.115, 95%CI: (0.019, 0.210)). Table 4 also shows that compared to mothers, fathers adopted more pressure to eat practices (*β* = 0.222, 95%CI: (0.153, 0.291)), but less monitoring (*β* = −0.220, 95%CI: (−0.305, −0.135)) practices.

## 4. Discussion

This study provided a brief insight into the health impact of parental coercive and structured feeding practices and their contributing factors among Chinese parents of preschoolers. Consistent with existing evidence [12], our results suggest that restriction and pressure to eat practices directly influence the children’s weights. This study highlighted the predictive value of parenting and family variables for parental coercive and structured feeding practices. Such findings may provide implications for future research studies and intervention decision-making to prevent obesity among younger children.

The findings indicate that the children of the parents who reported having used more restrictions and less pressure to eat practices currently had higher BMI levels, which is consistent with previous literature and supports the results from other populations [38,39,40,41]. However, some studies still failed to find the relationship between restriction practice and children’s weight [7,42], and some revealed a positive association between pressure to eat and children’s weights [5]. In addition, Gregory at al. indicted that the maternal feeding practice did not prospectively predict children’s BMI statuses, although it appears to influence the children’s eating performances [43]. Considering these mixed results, additional studies using a standard definition and measurement of coercive and structured feeding practices in a more diverse population of younger children are required to reaffirm the exact relationships and establish the potential dynamics among Chinese preschoolers. 

We observed that the phenomenon of parents’ incorrectly perceived children’s weight status widely existed among Chinese parents of preschoolers. In total, only 77.4% of paired parents correctly perceived the children’s weight, which was higher than the prevalence reported by Min et al. five years ago (68.3%) [44]. This prevalence is also slightly higher than studies from the US (73.7%) and the WHO European region, including 22 countries (64.1%) [45,46]. These might indicate that Chinese parents have learnt how to correctly identify their child’s weight. However, the parents of overweight/obese children have a higher misperception rate of their children’s weight than parents of children of normal-weight or underweight status in China, which is consistent with previous studies [19,47]. For overweight/obese children, 42.1% of parents underestimated their children’s level of overweight/obesity, although it is lower than the prevalence reported in two meta-analyses of international studies (50.7% and 55.0%, respectively) [48,49]. As in China, being fat means affluence. Chinese parents were glad to have their child chubbier, as they did not consider it a big issue for their child to be overweight and thus were more likely to underestimate their child’s overweight or obese weight status. Linear regression analysis indicated that parents’ incorrect perceptions of their children’s weight status only significantly influenced the pressure to eat practice in normal-weight children. For parents of normal-weight children, pressure to eat was positively associated with their underestimation of children’s weight and negatively associated with parents’ overestimation. Previous studies also demonstrated that when mothers perceived their normal-weight daughters to be underweight, they were more likely to pressure daughters to eat and restrict the foods available when they perceived their daughters to be overweight [50,51]. Belief and confidence factors were reported to mediate in the association between the weight perception and implementation of behaviors [52]. Chinese parents of overweight or obese children may lack the intention to improve their child’s weight or do not believe that they can change it through efforts, for which the associations of perceptions and feeding practices were only significant in normal-weight children.

Our results also showed that with the increasing BMI score of children, more parents have started to become concerned about their child being overweight. It suggests that parents were generally aware of their children’s weights and may be motivated to take action. However, according to the multivariate results, parents’ concerns about their child being overweight were associated with more restriction, monitoring, and less pressure to eat practices. Such adoption of practices may cause an additional increase in children’s weights. In this case, concerns about a child being overweight and the adoption of feeding practices will fall into a vicious cycle. The findings related to the perceptions of and concerns about children’s weight status were also found to be correlated with restriction, pressure to eat, and monitoring practices in the US [53]. Additionally, previous studies also suggested that childhood obesity interventions should address parents’ weight perceptions and concerns, considering the fact that a high proportion of parents have the misconception of a healthy weight and do not seek guidance from experts [54,55]. Therefore, our findings, paired with previous studies, suggest that the correct perceptions of children’s weights and appropriate concerns are essential for developing a healthy adoption of feeding practices and managing children’s body weights. Additionally, there is a need to guide the parents of Chinese preschoolers to estimate children’s weights correctly.

In the regression analysis of parenting stress and feeding practices, we observed that parenting stress was significantly associated with coercive feeding practices. Parents who reported higher levels of parenting stress used more restriction and pressure to eat practices. Stressed parents were more likely to use such feeding practices to compensate for the lack of control or insecurity they felt about the way they were able to care for their children [56]. Chinese parents perceived more stress related to rearing their children due to a high level of social competition; thus, they have a strong urge to ensure that their children eat what they need. Therefore, parents often use restriction and pressure to eat practices to guarantee their children’s health, although these feeding practices may oppose what they expected. In addition, parents who experience more parenting stress were less able to encourage a balanced diet because they were preoccupied with other responsibilities or stressors. Considering that, in China, we are trying to encourage parents to pay more attention to their children’s eating health and physical activities to prevent early childhood obesity, also concentrating on their children’s schoolwork and artistic improvements.

Our results also showed that a higher parenting competence was associated with more restriction, pressure to eat, and monitoring practices. Emma et al. [29] observed that higher maternal self-efficacy (MSE) decreased the instances of food restriction in a Western population, which is different from our findings. One explanation for the difference may be that we calculated a mean score of the four dimensions of parenting competence to explore the relationships in a Chinese population, while Emma et al. elucidated the associations between each dimension of competence and feeding practices in a Western population. An alternative explanation is that Chinese parents who perceive higher levels of parenting competence will be more confident about their traditional food parenting practices. Family and friends advised and supported such food parenting practices without understanding the potential impact on children’s health in the radically changed contemporary food supply [57,58]. Chinese parents traditionally prefer strict parenting practices, which may lead to positive associations between parenting competence and coercive feeding practices. However, since this study area is inadequately studied in China, there is a lack of evidence to support our findings. Therefore, further investigation is required to understand the relationships.

Our findings conclude that only child status may be a risk factor for preschoolers being overweight and obese. Regression analysis results showed that only child status of children was associated with parents using higher levels of pressure to eat, which may be an underlying process for increasing the child’s weight. Previous studies suggested that family mealtime interactions may mediate the association between only child status and a greater likelihood of overweight/obesity; the number of siblings and birth order may also influence the child’s eating habits [31,59]. We also observed that families where mothers or mothers and fathers were from one-child families had a stronger tendency to adopt restriction or monitoring practices. However, these potential pathways were not well established, primarily due to a lack of comprehensive behavior measures in previous studies. Thus, more studies need to explain the association and the potential pathways to understand how the only child status influences parental feeding practices. 

Our results show that fathers were more likely to use pressure to eat practices and less likely to monitor their children’s eating, which was significantly different from mothers’ feeding practices. Previous studies also suggested that fathers were more likely to adopt coercive feeding practices but less structural feeding practices [60]. Therefore, considering that many previous studies have focused on maternal influences and have ignored the fathers’ assignable roles in taking care of children in China, future studies and interventions should consider the impact of father–child feeding interactions.

Our study was among the first to examine the relationships between parenting stress, parenting competence, and coercive feeding practices in Chinese parents of preschoolers. This age group has far-reaching significance in the literature and is unique in its developmental expectations of increasing weight-related decision-making autonomy. Furthermore, our study had a large sample size, providing adequate power to detect the effects outlined in our hypotheses. Despite these strengths, several limitations should be noted. First, the cross-sectional nature of this study design limited our ability to determine causal relationships. Second, feeding practices were evaluated using a single parent-reported questionnaire without direct observation and measurement. Third, our measure of parenting relied on a questionnaire and combined all dimensions of parenting stress and competence into one score without considering that different relationships may exist in every dimension. Fourth, our study’s limited district diversity also represented a limitation of generalization. Fifth, given that this sample only examined the primary associations between parenting, family characteristics, and feeding practices, future work exploring the potential pathways of these links is also essential. Finally, paired with previous literature examining the indicators of parental feeding practices, our research findings justify future longitudinal, mixed-methods research, which can more robustly measure parental factors and their impact on parent–child feeding interactions and children’s health later in life.

## 5. Conclusions

Our study demonstrated that parents’ perceptions and concerns about weight, parenting stress, parenting competence, and only child status were significantly associated with restriction, pressure to eat, and monitoring of feeding practices. Fathers’ feeding preferences were significantly different from what mothers preferred. Our study results provide a framework for researchers and practitioners that, when designing family-based programs for childhood obesity prevention, family factors, general parenting perception, and parental family roles are essential. This may help to lower childhood obesity rates.

## Figures and Tables

**Figure 1 nutrients-14-03109-f001:**
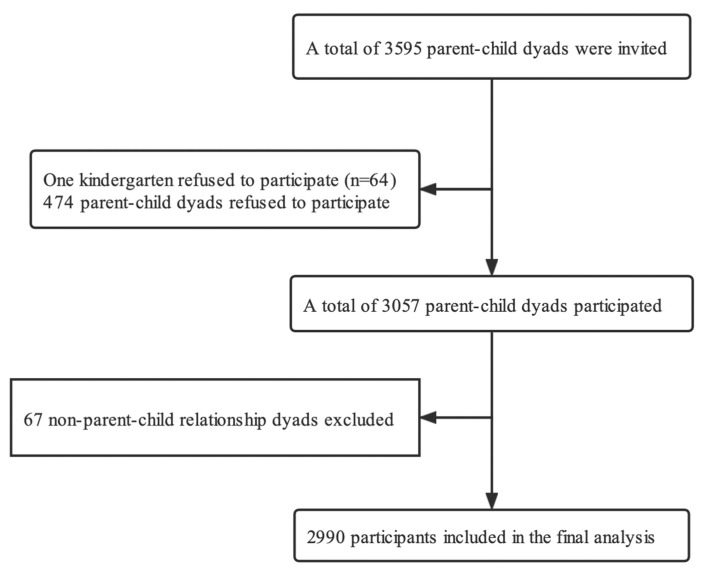
Flow chart of the section process of the participants.

**Figure 2 nutrients-14-03109-f002:**
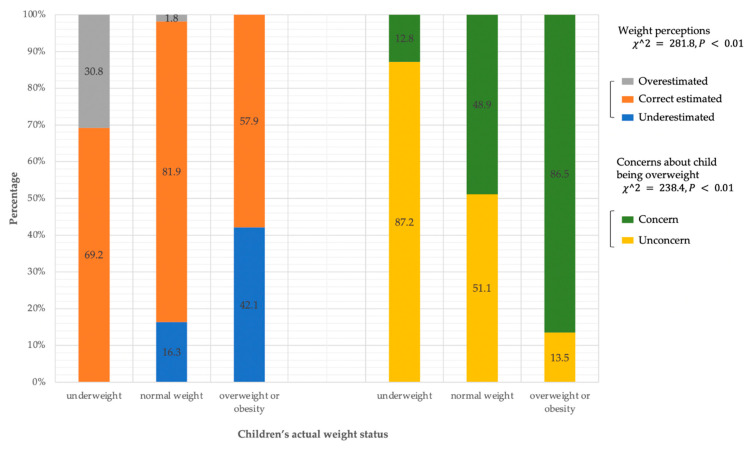
Parental weight perception, concerns, and actual weight status of preschoolers.

**Table 1 nutrients-14-03109-t001:** Characteristics of the preschoolers and parents: *n* = 2990.

Characteristic	*n*	%
Children’s characteristics		
Gender		
Male	1546	51.7
Female	1444	48.3
Age (Month)		
Mean (SD)	66.3	7.3
Median (Min, Max)	65.0	[33.0, 84.0]
Only child status		
Yes	2059	68.9
No	931	31.1
BMI		
Mean (SD)	15.6	1.8
Median (Min, Max)	15.3	[10.6, 34.7]
Weight status		
Normal weight	2419	80.9
Underweight	39	1.3
Overweight	363	12.1
Obesity	169	5.7
Parents’ characteristics		
Family roles		
Father	688	23.0
Mother	2302	77.0
Marital status		
Married	2851	95.4
Others	139	4.6
Household income per capita (CNY)		
≤80,000	261	8.7
81,000–100,000	386	12.9
101,000–150,000	591	19.8
>150,000	1605	53.7
From a one-child family		
Father	466	15.6
Mother	554	18.5
Both	1333	44.6
None	637	21.3
Perception of children’s weight		
Correct perception	2315	77.4
Misperception	675	22.6
Parenting stress		
Mean (SD)	2.3	0.8
Median (Min, Max)	2.2	[1.0, 5.0]
Parenting competence		
Mean (SD)	4.0	0.6
Median (Min, Max)	4.0	[1.0, 5.0]

Note: SD, standard deviation; BMI, body mass index; CNY, Chinese Yuan; USD, United States dollar.

**Table 2 nutrients-14-03109-t002:** Scores of feeding practices according to children’s weight status: Mean (SD).

Feeding Practices	Children’s Actual Weight Status				Total (*n* = 2990)		*p* Value
	Normal Weight (*n* = 2419)	Underweight (*n* = 39)	Overweight (*n* = 363)	Obesity (*n* = 169)	Mean (SD)	Median [Min, Max]	
Restriction	3.8 (0.8) ^a^	3.7 (0.8) ^ab^	3.9 (0.7) ^b^	4.0 (0.8) ^b^	3.8 (0.8)	4.0 [1.0, 5.0]	**<0.001**
Pressure to eat	3.3 (0.8) ^a^	3.7 (0.6) ^b^	3.1 (0.8) ^c^	2.9 (0.9) ^c^	3.2 (0.8)	3.0 [1.0, 5.0]	**<0.001**
Monitoring	3.9 (1.0)	3.9 (1.0)	3.9 (1.0)	4.0 (1.0)	3.9 (1.0)	4.0 [1.0, 5.0]	0.215

Note: SD, standard deviation, *p*-values were calculated using Welch’s ANOVA test and analysis of variance, respectively. Bold indicates statistically significant values. Dunnett-T3 were calculated for post hoc analysis. The same letter on the same row of data indicates no significance *p* > 0.05.

**Table 3 nutrients-14-03109-t003:** Linear regression relationships between parents’ weight perceptions, concerns, and feeding practices among different weight statuses of children.

	Restriction *β* (95%CI)	Pressure to Eat *β* (95%CI)	Monitoring *β* (95%CI)
Normal weight (*n* = 2419)			
Misperception of child’s weight			
Correct estimate	1	1	1
Underestimate	0.028 (−0.060, 0.117)	**0.333 (0.243, 0.423) ****	−0.063 (−0.174, 0.049)
Overestimate	0.136 (−0.106, 0.379)	**−0.272 (−0.518, −0.026) ***	−0.259 (−0.564, 0.046)
Concern about their child being overweight			
Unconcerned	1	1	1
Concerned	**0.207 (0.140, 0.273) ****	**−0.078 (−0.145, −0.011) ***	**0.129 (0.046, 0.213) ****
Underweight (*n* = 39)			
Misperception of child’s weight			
Correct estimate	1	1	1
Overestimate	0.074 (−0.672, 0.819)	−0.290 (−0.767, 0.186)	0.112 (−0.863, 1.087)
Concern about their child being overweight			
Unconcerned	1	1	1
Concerned	−0.168 (−1.147, 0.811)	−0.415 (−1.041, 0.211)	−0.069 (−1.350, 1.212)
Overweight or obesity (*n* = 532)			
Misperception of child’s weight			
Correct estimate	1	1	1
Overestimate	−0.012 (−0.140, 0.116)	0.122 (−0.022, 0.266)	−0.087 (−0.264, 0.089)
Concern about their child being overweight			
Unconcerned	1	1	1
Concerned	**0.501 (0.316, 0.686) ****	0.103 (−0.105, 0.311)	**0.356 (0.102, 0.611) ****

Note: CI, confidence interval, multiple linear regression analysis. Bold indicates statistically significant values, * *p* < 0.05, ** *p* < 0.01. Regression model with adjustment for child gender, age, parental family role, parenting, marital status, household income, and only child status of both parent and child.

**Table 4 nutrients-14-03109-t004:** Linear regression relationships between family characteristics and feeding practices.

	Restriction *β* (95%CI)	Pressure to Eat *β* (95%CI)	Monitoring *β* (95%CI)
Parenting stress and competence			
Parenting stress	**0.048 (0.010, 0.087) ***	**0.159 (0.120, 0.198) ****	−0.026 (−0.074, 0.022)
Parenting competence	**0.057 (0.084, 0.105) ***	**0.060 (0.010, 0.109) ***	**0.253 (0.193, 0.314) ****
Only child status for children			
Yes	1	1	1
No	0.007 (−0.055, 0.069)	**0.108 (0.044, 0.171) ****	0.000 (−0.078, 0.078)
From a one-child family			
None	1	1	1
Father	−0.041 (−0.137, 0.054)	−0.071 (−0.169, 0.027)	0.011 (−0.110, 0.106)
Mother	**−0.109 (−0.200, −0.019) ***	**−0.097 (−0.190, −0.004) ***	0.108 (−0.007, 0.222)
Both	**−0.096 (−0.171, −0.020) ***	−0.058 (−0.135, 0.019)	**0.115 (0.019, 0.210) ***
Parental family role			
Mother	1	1	1
Father	0.038 (−0.030, 0.105)	**0.222 (0.153, 0.291) ****	**−0.220 (−0.305, −0.135) ****

Note: CI, confidence interval, multiple linear regression analysis. Bold indicates statistically significant values, * *p* < 0.05, ** *p* < 0.01. Regression model with adjustment for child gender, age, BMI, marital status, and household income.

## Data Availability

According to private and confidential clauses stated in the informed consent, the dataset generated and analyzed during the current study is ethically restricted and not publicly available.

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
