# Peer review of "How Parenting and Family Characteristics Predict the Use of Feeding Practices among Parents of Preschoolers: A Cross-Sectional Study in Beijing, China"

_nutrients, 2022, doi:10.3390/nu14153109_

Round 1

Reviewer 1 Report

The manuscript entitled “ How Parenting and Family Characteristics Predict the Use of Feeding Practices among Parents of Preschoolers: A Cross-Sectional Study in Beijing, China” presented few but interesting data. The major issue with this paper is that it does not relate their findings to other studies around the world such as in Southern Europe where the occupation of the mother or fathers’ exercise status played a significant role in childhood obesity. The fact that parents of overweight or obese children have the misconception of healthy weight and do not seek guidance from obesity experts early on has also been reported . Thus,it would be scientifically sound to compare and contrast your findings to other similar in design studies. Moreover, stress is a very interesting and important factor in obesity prevalence. The manuscript would greatly benefit if there was a critical evaluation of the findings along with stress scientific data, not only related to emotional eating.

Reviewer 2 Report

The study by Hu et al. is a well-designed and described prospective study aimed to explore associations between parental and family characteristics of children feeding practices. The authors conducted a study on 2990 parents of preschoolers. I have several comments:

1.       Authors conducted ANOVA analysis to analyze the differences between groups, however, they did not conduct an analysis of the normality of distribution and homogeneity of variance. Moreover, they did not a post hoc analysis.

2.       Table 1: I suggest adding the min and max values into quantitative variables. Please, explain the RMB abbreviation.

3.       Table 2: Please add the min and max values and results of the post hoc analysis

4.       Figure 2: placing of the title of this figure should be corrected. Moreover, placing results of the analysis regarding weight perception and concerns about weight separately on the two sides of the figure would be more readable than mixing them and presenting regarding weight status. Additionally, the results of Chi2 should be added into the figure. Also, how big were groups if in the whole group only 39 kids were underweight (this should also be added in the table 3)?

5.       In my opinion, another linear regression model should be built, including parents' perception and concerns and family characteristics.

6.       Discussion should also include insight into children eating behaviors, as well as their associations with feeding practices e.g.: https://doi.org/10.1186/1479-5868-7-55, DOI: 10.1111/j.1747-0080.2012.01631.x,  https://doi.org/10.3390/nu14112279, https://pubmed.ncbi.nlm.nih.gov/23110748/, DOI: 10.1016/j.appet.2010.02.013

7.       References should be corrected according to the instructions for authors: https://doi.org/10.1186/1479-5868-7-55, DOI: 10.1111/j.1747-0080.2012.01631.x,  https://doi.org/10.3390/nu14112279, https://pubmed.ncbi.nlm.nih.gov/23110748/, DOI: 10.1016/j.appet.2010.02.013

8.       References should be corrected according to the instructions for the authors
